# The Tapeworm *Hymenolepis diminuta* as an Important Model Organism in the Experimental Parasitology of the 21st Century

**DOI:** 10.3390/pathogens11121439

**Published:** 2022-11-29

**Authors:** Anna Sulima-Celińska, Alicja Kalinowska, Daniel Młocicki

**Affiliations:** 1Department of General Biology and Parasitology, Medical University of Warsaw, Chałubińskiego 5, 02-004 Warsaw, Poland; 2W. Stefański Institute of Parasitology, Polish Academy of Science, Twarda 51/55, 00-818 Warsaw, Poland

**Keywords:** *Hymenolepis diminuta*, experimental parasitology, helminth therapy, host–parasite interrelationship, immunology, model organism, molecular biology, tapeworm

## Abstract

The tapeworm *Hymenolepis diminuta* is a common parasite of the small intestine in rodents but it can also infect humans. Due to its characteristics and ease of maintenance in the laboratory, *H. diminuta* is also an important model species in studies of cestodiasis, including the search for new drugs, treatments, diagnostics and biochemical processes, as well as its host–parasite interrelationships. A great deal of attention has been devoted to the immune response caused by *H. diminuta* in the host, and several studies indicate that infection with *H. diminuta* can reduce the severity of concomitant disease. Here, we present a critical review of the experimental research conducted with the use of *H. diminuta* as a model organism for over more than two decades (in the 21st century). The present review evaluates the tapeworm *H. diminuta* as a model organism for studying the molecular biology, biochemistry and immunology aspects of parasitology, as well as certain clinical applications. It also systematizes the latest research on this species. Its findings may contribute to a better understanding of the biology of tapeworms and their adaptation to parasitism, including complex correlations between *H. diminuta* and invertebrate and vertebrate hosts. It places particular emphasis on its value for the further development of modern experimental parasitology.

## 1. Introduction

For ethical reasons, research on human parasites must be carried out in model systems. The species used in such models are a small group of research organisms that are convenient for studying the mechanisms regulating the functioning of living organisms. Initially, the simplest organisms such as bacteria or bacteriophages were used for research. With progress, research has included more complex organisms, such as *Drosophila*, zebrafish or *Caenorhabditis elegans*. As some of these models share similar aspects of biology with us, some findings can also be applied to the human body. Model organisms have thus become irreplaceable tools in fundamental biological and clinical research [1].

Cestodology is a section of parasitology that studies tapeworms, and has employed various model species. One such species is *Echinococcus multilocularis*, which has been described as a model of the interplay between parasite and intermediate host [2]. However, as the adult stage of *Echinococcus* spp. requires the use of large vertebrate hosts, development and testing in vivo is difficult in the laboratory. Another parasite used as a model organism is *Hymenolepis microstoma*. Fortunately, the genome of *H. microstoma* is well studied [3], and the species can be used as a classical tapeworm model for research in the genomic era. Although *H. microstoma* is also used to study the development and evolution of tapeworm segmentation [4], it only survives for around six months in the host organism and requires frequent passage. Another experimental model is *Mesocestoides corti* [5], which is used for studies of cestode biology and its transition from the larval tetrathyridium to the strobilated stage [6]. However, while *M. corti* develops in vivo only in the tetrathyridium stage, all strobilated stages are strictly in vitro.

From a scientific point of view, the best model organism for research on cestodiasis is *H. diminuta*. This parasite has a worldwide distribution as an intestinal parasite. It occurs in three stages: egg with oncosphere, cysticercoid and adult (Figure 1). Eggs are excreted with the faeces of the definitive host, and these may serve as a source of infection for the intermediate host (beetles *Tenebrio molitor* or *Tribolium castaneum*). After being eaten, the eggs with oncospheres develop into the cysticercoid stage in the intermediate host. The cysticercoid is an invasive stage for the definitive host, which gets to the rat or human body by consumption of the infected insects, either directly or via contaminated water or food. In the body of the definitive host (rodents or humans), the cysticercoid transforms into the adult stage and lives in the small intestine as a parasite. The entire lifecycle of *H. diminuta* can be easily performed in the laboratory and in vivo culture can be performed using natural hosts. *H. diminuta* is the only such species that fulfills the conditions necessary to become a model organism for cestodiasis.

The advantages of using *H. diminuta* as a model tapeworm parasite in laboratory studies are summarized by McKay [7]. Firstly, the tapeworm is not auto-infective; there is no risk of the parasite spreading among animals when no arthropod is present. Moreover, *H. diminuta* has no structures which could damage the small intestine, and is a non-migratory and non-invasive parasite. Furthermore, *H. diminuta* is easy to clear from the host body by anthelminthic treatment [7]. Thus, *H. diminuta* is easy to grow in laboratory conditions and has the potential to enable research based on modern molecular techniques [8]. For example, a beetle–rat tapeworm (*H. diminuta*) model is utilized to evaluate the efficacy of known anthelmintic compounds (praziquantel, mebendazole) against *H. diminuta* cysticercoid in vitro [9]. The tapeworm shares the same basic developmental features with most other cestodes; as such, the study of *H. diminuta* has significantly contributed to a better understanding of cestodes in general, including their development, reproduction and survival strategies [10].

This review focuses on the use of *H. diminuta* as a model species in research on cestodiasis, including attempts to identify new drugs and methods of treatment, diagnostics, biochemical processes, immune responses and other interactions in the parasite–host system (Figure 2). Based on a review of nearly 140 publications completed over 22 years, the present study systematizes current knowledge about the biochemistry, immunology, host–parasite relationships and molecular biology of *H. diminuta* parasitoses and their treatment.

## 2. Molecular Biology

The rapid development of proteomic and other molecular techniques over recent decades has yielded modern research tools which are able to provide greater insight into a range of pathologies in animals and humans, including parasitoses. Today, the molecular biologist has a huge number of research methods at her disposal. In the beginning of the 21st century, a study based on SDS-PAGE, Western blot and RT-PCR identified a protein that could act as a new diagnostic tool for tapeworm infection. The sequence analysis indicated that the protein was a member of the hydrophobic ligand binding protein family. Later, the biological function of this protein was determined with the use of a rat-tapeworm *H. diminuta* model [11].

PCR variants are very popular in the study of cestodiasis. For example, von Nickisch-Rosenegk et al. (2001) determined the complete 13,900-nt sequence of the mitochondrial genome of *H. diminuta* using long-PCR. The authors observed 13 repeats of a 31-nt sequence and a potential stem-loop structure of 25 bp with an 11-member loop. The findings allowed *H. diminuta* to be placed in the Eutrochozoa [12]. In turn, Nowak et al. (2019) attempted hybrid de novo whole-genome assembly of the *H. diminuta* based on complementary sequencing technologies and methods. The final genome assembly was found to extend for 177 Mbp and the authors annotated 15,169 potential protein-coding genes. The initial findings from the hybrid de novo whole-genome sequencing of *H. diminuta* combined with RNAseq analysis provided a better understanding of the biology of the parasite; in addition, the hybrid sequencing method made it possible to obtain high-quality data [13]. This is particularly important from the point of view of modern research on the evolution and phylogeny of organisms, where the analysis of genes plays a key role. In addition, understanding the genomes allows for detailed functional analyses and diversity within genes and their families.

Other studies focused more on genes. Mohajer-Maghari et al. (2007) cloned, sequenced and characterized the *H. diminuta* alpha-tubulin gene. By constructing a lambda phage cDNA library, it was possible to isolate a full-length alpha-tubulin cDNA. The *H. diminuta* alpha-tubulin gene comprised approximately 1400 bp and consisted of 450 amino acids. The alpha-tubulin isolated from *H. diminuta* was found to possess 10 distinctive residues, which are not found in any other α-tubulins; it is possible that the protein may allow the parasite to survive within the host [14]. This may point to new specific therapeutic targets having practical implications. In turn, Řežábková et al. (2019) explored the diversity of laboratory-kept strains using molecular-phylogenetic analyses. A comparison of the lsrDNA- and cox1-sequences of nine isolates of *H. diminuta* with available sequences of *H. diminuta* deposited in GenBank found that very little genetic diversity was present in *H. diminuta* isolates. These results proved that the *H. diminuta* used in medical research may originate from a limited number of sources [15]. These observations have the advantage that they indirectly indicate high reproducibility of test results in different laboratories. Perhaps the results of comprehensive studies conducted with *H. diminuta* would not be so reproducible if the genetic differences were significant.

Another study used PCR and RFLP of the rDNA-ITS2 gene marker to distinguish *H. diminuta* from *H. nana*. Although both *Hymenolepis* species’ rDNA had ITS2 regions comprising 800 bp, they differed in the digestion patterns. Also, the ITS2 sequences found differences in secondary structure and in length between the species: 586 bp in *H. diminuta* and 531 bp in *H. nana* [16]. PCR-RFLP was useful for differentiating between the two species at the molecular level. Hoque et al. (2007) examined the suitability of house-keeping genes to serve as internal controls for intestinal tissues in rats infected with *H. diminuta*. Briefly, RNA was isolated from the samples and cDNA was synthesized from the templates. The authors analyzed the mRNA levels of four house-keeping genes (GAPDH, β-actin, 18S and HPRT) and Muc2 using semi-quantitative RT-PCR. The results indicate that there is no obvious house-keeping gene to use as a standard for normalization in an in vivo study in *H. diminuta* infected rats [17].

Other studies used quantitative real-time PCR (qRT-PCR) to measure the changes in the expression of TLR3 and TLR9 [18], as well as TLR2 and TLR4 [19,20] in the jejunum and colon of rats, with TLR3 and TLR9 expression being further quantified using Western blotting [18]. Kosik-Bogacka et al. (2014) found TLR3 gene expression to increase significantly only at 16 days post-infection (dpi) and TLR9 expression to increase markedly at 16 and 25 dpi in the jejunum. In turn, TLR3 and TLR9 expression was significantly upregulated at 16, 25, 40 dpi in the colon. These results imply that TLR3 and TLR9 took part in the immune response against the invasion of *H. diminuta* [18]. A detailed study found that rats infected with *H. diminuta* demonstrated strong expression of mRNA for TLR4 and TLR2 in the jejunum compared with uninfected controls. TLR2 expression increased in the rat jejunum 6 and 8 dpi and in the rat colon at 8 dpi. In turn, TLR4 expression increased in the rat jejunum at 4, 6 and 8 dpi and in the colon at 6 dpi. The authors suggest that the increase in TLR expression may have been induced by heat shock proteins [19]. In addition, at 16 dpi, the expression of TLR4 and TLR2 in the jejunum was higher in infected rats than uninfected rats. In turn, authors observed increased expression of TLR2 from 16 to 40 dpi and TLR4 from 16 to 60 dpi in the colon. Higher numbers of intestinal bacteria were also noted in *H. diminuta* infected rats. However, microbiota had only a small effect on the expression of TLR2 and TLR4. Despite this, the studied TLRs and the changes in their expression played a role in the pathomechanism of *H. diminuta* infection [20]. COX-1 and COX-2 expression was also evaluated using Western blotting in rats infected with *H. diminuta*. It was found that *H. diminuta* infection increased COX-1 and COX-2 protein expression in the jejunum and colon of infected rats. The presence of the parasite caused inflammation depending on the expression and activation of cyclooxygenases [21].

Further research should lead to elucidation of the mechanisms of changes in receptor expression and indicate whether, and which, parasitic proteins may be involved in this process. It is possible that tapeworms may affect intestinal tissue indirectly, e.g., by affecting the microbiome, which in turn may lead to changes in the expression of receptors. These hypotheses may be verified in the near future.

A qRT-PCR study examined the effect of *H. diminuta* infection on 29 immunity-related genes exhibiting antimicrobial properties in two strains of *Tribolium castaneum*: one resistant and another susceptible. Before exposure, 13 genes demonstrated different levels of exposure between strains; however, after exposure, nine genes were expressed in the susceptible strain and 13 genes in the resistant strain. Seven genes demonstrated 10-fold higher expression in the resistant than the susceptible strain. These findings confirm the influence of the genetic background on the regulation of immunity-related gene expression [22].

A study based on a combination of PCR, Southern blot and Northern blot demonstrated that the *H. diminuta* molecule influences the reproduction of beetles (*T. molitor*) by manipulating the expression of vitellogenin genes [23]. In contrast, a study of muscarinic receptor mRNA in the jejunum of rats based on semiquantitative RT–PCR found that the presence of large numbers of worms suppresses the relative expression of M1 receptor mRNA. The results show that low-level infections with rat tapeworms leads to changes in enteric cholinergic activity and may reflect neural functioning [24].

The AFLP (amplified fragment length polymorphism) technique is based on selective PCR amplification of restriction fragments from a total digest of genomic DNA. The technique was used by Zhong et al. (2003) to examine the genetic basis of susceptibility to *H. diminuta* in *T. castaneum*. Briefly, quantitative trait loci (QTL) mapping experiments were conducted using AFLP markers and randomly-amplified polymorphic DNA (RAPD) markers. The results suggest that the polygenic nature of the beetle and epistasis play key roles in susceptibility to tapeworm infection [25]. Zhong et al. (2005) also examined whether host resistance to parasite infection is associated with fitness costs; a QTL scan using AFLP markers revealed three QTLs for beetle resistance to tapeworm infection, five QTLs for fecundity and another five QTLs for egg-to-adult viability. The three QTL conferring resistance were colocalized with the QTL affecting beetle fitness. The findings indicate that the multiple loci conferring resistance to *H. diminuta* infection were partially responsible for fitness costs in the resistant beetle populations [26]. Conducting research on invertebrates is much easier and does not raise ethical concerns. Many of the observations made may have general implications and similar mechanisms can then be studied and sought or confirmed in mammalian models.

In addition, Warr et al. (2004) used a combination of acridine orange and TUNEL (terminal-deoxynucleotidyl-transferase-mediated dUTP-biotin nick end labelling) to examine apoptosis in the fat body tissue of the beetle *T. molitor* parasitized by *H. diminuta*. The aim of the study was to investigate the mechanisms underlying the reduction in fecundity in the parasitized *T. molitor* in vivo. The findings indicate that parasite infection could indirectly induce apoptosis, but the induction signal does not originate directly from *H. diminuta* [27]. It would also be interesting to see if *H. diminuta* can indirectly affect the process of apoptosis, fat metabolism and possibly fecundity in vertebrates.

Due to the use of *H. diminuta* in molecular studies, it was possible to identify and characterize genes influencing the tapeworm’s evolutionary success. Moreover, the whole genome has been described which is crucial for identifying individual genes/proteins contributing to adaptation to environmental constraints and host–parasite interactions. Data obtained from genome sequencing are useful for the understanding of tapeworm biology, development, their adaptations to the parasitic way of life and host–parasite physiological relationships in vertebrate and invertebrate hosts. The identification and characterization of specific genes may also help to find new diagnostic and treatment methods in the future.

## 3. Biochemistry

*H. diminuta* is used as a model species in a number of biochemical research. A biochemical characterization of HLBP (hydrophobic ligand binding protein) from *H. diminuta* [28] found that HLBP had a single tryptophan at position 41 [11], and that this was involved in ligand binding [28]. A more detailed examination showed that HLBP bound a range of anthelmintics. Such interactions between anthelmintics and HLBPs may be important in determining drug specificity. The presence of high HLBP levels in parasitic helminths suggests that these hydrophobic binding proteins play an important role in metabolism [28]. Onufriev et al. (2005) examined the effect of a nitric oxide donor (NOD) on the synthesis of cGMP in adult *H. diminuta* by radiometric assay. A high level of cGMP was observed in the nerve fibers of the tapeworm, which are closely associated with muscle fibers. This indicates that NO takes part in motor neuron activation [29].

Park and Fioravanti (2006) found NADH→NADP + transhydrogenation to be both a non-energy-linked and energy-linked process [30]. During NADPH and NAD generation in *H. diminuta*, proton-translocating mitochondrial transhydrogenase can serve as an additional anaerobic phosphorylation site [31]. A study of the proteolytic activity of egg capsules identified a gelatinase-type proteolytic enzyme which was present at the time of capsule formation; the activity of the enzyme was inhibited by leupeptin but not by EDTA or PMSF, suggesting it may be a cysteine proteinase [32].

Skrzycki et al. (2011) examined the potential of the antioxidant system in *H. diminuta* for suppressing the host’s defense. Oxidative stress was determined by the concentrations of thiobarbiturate reactive substances (TBARS) and reduced glutathione (GSH), and the antioxidant enzyme activity of superoxide dismutase (SOD1 and SOD2), catalase, glutathione peroxidases, glutathione transferase and glutathione reductase. The results suggest that tapeworms reacted to increased ROS levels by increasing the activity of antioxidant enzymes [33]. In turn, Czeczot et al. (2013) report elevated lipid peroxidation but decreased GSH concentration and SOD activity in *H. diminuta* in particular parts of the small intestine of rats. These findings indicate that the parasite employs defense strategies against oxidative stress. Chronic invasion of the rat by tapeworms led to a weakening of the antioxidant defense mechanisms [34].

Kosik-Bogacka et al. (2011) studied the intensity of oxidative stress and the effectiveness of antioxidant protection during *H. diminuta* invasion in rats. The rats demonstrated changes in lipid peroxidation and glutathione levels, and alterations in the activity of antioxidant enzymes (superoxide dismutase, catalase, glutathione reductase and glutathione peroxidase). In addition, significant decreases in superoxide dismutase activity and increases in lipid peroxidation products were noted in the duodenum and jejunum of the infected rats. The results indicate that the presence of the parasite may influence the balance between oxidant and antioxidant processes [35].

A study based on UHPLC/MS/MS found that *H. diminuta* enzymatic systems are not able to metabolize praziquantel (PZQ) [36]. Another study of the metabolites of anthelmintics using HPLC with spectrofluorometric or mass-spectrometric detection found that *H. diminuta* produces enzymes that are able to ameliorate the toxic effect of flubendazole and mebendazole by reducing their carbonyl groups [37]. Kreshchenko et al. (2021) confirmed the presence of 5-HT7 type serotonin receptors, but not 5-HT, in the flame cells of *H. diminuta* larvae. This may indicate its involvement in the regulation of the contractile activity of the flame cells, and thus the locomotor system. The results suggest that the serotoninergic nervous system of *H. diminuta* could be a possible target for anti-parasite drugs [38].

The above studies indicate various metabolic and physiological aspects of tapeworm biology, the explanation of which may also have practical aspects and contribute to the development of new therapies or understanding the mechanisms influencing the parasite response to oxidative stress.

Understanding the mechanisms involved in the regulation of the oxidative balance by tapeworms would allow us to discover new mechanisms of parasitic immunomodulation, especially that the processes taking place in host–parasite immunology are complex and still poorly understood.

## 4. The Host–Parasite Immunology

The host immune system is designed to protect against foreign organisms, including parasites, and hence affects parasite abundance and survival. As such, parasites have developed mechanisms that ensure a suitable environment for their existence and survival in the host body by evading or manipulating the host immune system. In rats, the presence of *H. diminuta* induces a type 2 immune response and an immunoregulatory [39] response, which is consistent with previous studies [7]; this is characteristic of helminth infections in general [40].

In a mouse host, the response follows the classical Th2 pattern associated with helminthic infections and is dominated by IL-4, IL-5 and IL-13 [7]. Infection of rats with large numbers of *H. diminuta* also stimulates a Th2 immune response, resulting in high levels of IL-13 [41]. Wang and McKay (2005) report that *H. diminuta* extract reduced IL-2 and IL-4, but enhance the production of IFNγ and IL-10. By inhibiting IL-4 and promoting IL-10 production, *H. diminuta* blocks stimulated T cell proliferation and may bias the immune environment towards immunoregulation and away from the Th2-type immune response [42]. The immunomodulatory properties of *H. diminuta* seem indisputable. It is known that the parasitic extract contains molecules with immunomodulatory properties. Identification of individual proteins with such activity would allow to understand the mechanisms of these processes, whereas obtaining recombinant proteins would allow us to check their therapeutic potential. This seems increasingly likely in the light of recent genomic and proteomic advances.

McKay and Khan (2003) examined the effect of *H. diminuta* tapeworm infections in mice with knockout IL-4, IL-13 and STAT-6 (signal transducer and activator of transcription 6) genes. Only the STAT-6 knockout mice became infected with tapeworms; no adult stage *H. diminuta* were observed in the IL-4 knockout or IL-13 knockout mice. The findings suggest that STAT-6 was necessary for the expulsion of *H. diminuta* from mice [43]. The mechanisms involved in parasite expulsion should also be investigated in other animal models. It is well known that only young rats became infected with *H. diminuta*. Similarly, most cases of human infection are recorded in children. This may indicate similar mechanisms of maturation in juvenile/mature rats and humans, and limit the possibility of using *H. diminuta* as a potentially therapeutic organism.

One mechanism facilitating the immunological response is the mucosal mast cell response, which may play a role in generating intestinal changes associated with the expulsion of intestinal tapeworms. Ishih and Uchikawa (2000) suggest that the tapeworm biomass was related to mast cell and IgE responses, but that these responses were not required for tapeworm expulsion during low dose *H. diminuta* infection in rats [44]. Rats with mast cell deficiency displayed reinfection resistance for tapeworm biomass but not tapeworm expulsion. The researchers suggest that the mast cell plays a role in controlling the biomass of *H. diminuta* in reinfection alone but does not affect the rate of tapeworm expulsion. Hence, it can be assumed that the mast cell is not a major effector cell for controlling the carrying capacity of tapeworms. A study based on rat strains carrying genetic mutations (T, B and NK cells deficiency; T cell deficiency and mast cell deficient) suggests that the host immune system can control maximal tapeworm biomass, i.e., carrying capacity, insofar that T cells, B cells and NKs appear to suppress carrying capacity of *H. diminuta* [45].

Most studies of the mammalian immune response to intestinal parasitic infections focus on the resistance of T helper cells (Th) [46]. For example, in a study of the role of enteric epithelial cells in response to infection with *H. diminuta,* Lopes et al. (2015) found that an intestinal epithelial cell layer promoted the in vitro survival of *H. diminuta* and determined host permissiveness to infection; briefly, rat tapeworm was cultured with rat, human or mouse epithelial cell lines, and while all promoted worm survival, those cultured on rat were the healthiest, as indicated by their mobility [47]. Starke-Buzetti and Oaks (2008) report a change in the number of cells containing the glial-derived neurotrophic factor (GDNF) in the jejunum and ileum of rats following *H. diminuta* infection. Their findings indicate that GDNF was present in infected rat macrophages, suggesting that GDNF may be involved in the response to parasitic infections of the intestine [48].

In parasitic organisms, a key role in invasion is played by excretory–secretory proteins (ESPs) or surface proteins, which act as virulence factors and immune regulators that control host immune recognition during infection. They are also crucial for ensuring parasite survival inside and outside the host, and their expressions undergo changes in response to environmental stimuli. It is commonly known that a number of ESPs are important sources of immunomodulatory and antigenic proteins, and are easily recognized by the host immune system. Bień et al. (2016) propose that ESPs may play a role in regulating host immunity and influencing the surrounding environment. Many of the proteins identified within ESP appear to be engaged in immunomodulatory processes, for example Heat Shock Proteins (HSP) [49]. Zawistowska-Deniziak et al. (2017) found that ES molecules of *H. diminuta* may contain anti-inflammatory and immunomodulatory proteins. Studies on human macrophage polarization by *H. diminuta* indicate that ESPs serve as better sources of anti-inflammatory molecules than somatic proteome. Macrophages treated with parasite ESPs demonstrated reduced expression of cytokines (i.e., IL-1α, TNFα, TGFβ, IL-10) and chemokines (i.e., IL-8, MIP-1α, RANTES, and IL-1ra). In addition, inflammatory factor expression rose significantly when macrophages were exposed to living parasites. Moreover, the authors suggest that exposure to *H. diminuta* antigens and ESPs induces a mixed type of macrophage polarization [50]. A similar study was conducted by Johnston et al. (2010), who assessed the ability of extract of adult *H. diminuta* to suppress macrophage activity; exposure to the high molecular-weight extract of *H. diminuta* was observed to reduce macrophage activation [51]. In addition, Aira et al. (2017) found antigens from the rat tapeworm to cause an anti-inflammatory response characterized by M2-type polarization, reduced macrophage phagosome maturation and a reduced ability to activate T cells [52].

Neutrophils are considered first-line responders to microbial infections, which generate proteases and ROS (reactive oxygen species) to kill pathogens. They can also play important roles in the host’s response following infection with helminth parasites. Graves et al. (2019) tested *H. diminuta* extract for its ability to activate murine neutrophils and affect neutrophil chemotaxis in vitro. Their findings indicate that neutrophils play important roles in the host response following infection with helminth parasites. A detailed examination showed that while *H. diminuta* extract is not a chemotactic stimulus for murine neutrophils, the proteoglycan in the extract significantly reduced neutrophil chemotaxis. The authors suggest that this extract can be used to modify the outcome of neutrophilic disease [53].

The use of *H. diminuta* made it possible to investigate parasite influence on host immune cells. The research presented by the scientists explains the mechanisms of immune resistance that are triggered by tapeworms. Not only infection with an adult *H. diminuta* tapeworm, but also tapeworm extract or ESPs have immunomodulatory properties. Parasite-derived proteins are a unique source of immunomodulatory molecules, but there is a need for more comprehensive studies on the interaction and role of these molecules during parasitic infections.

## 5. *H. diminuta* as a Therapeutic Agent

A number of studies have shown that infection with *H. diminuta* can reduce the severity of concomitant disease. The effect of *H. diminuta* infection has been studied on dextran–sodium sulphate (DSS)-induced colitis, dinitrobenzene sulphonic acid (DNBS)-induced colitis, oxazolone-induced colitis, acetic acid-induced gastric ulceration and Freund’s complete adjuvant (FCA)-induced arthritis. Rats are permissive hosts for *H. diminuta*, but mice as non-permissive hosts eradicate a primary infection of tapeworm within 14 days. This feature allows for the comparison of the impact of *H. diminuta* infection on the course of colitis induced in permissive and non-permissive hosts [7].

Hunter et al. (2005) demonstrated that *H. diminuta* infection reduces the severity of DNBS-induced colitis in mice, both before and after treatment; however, the protective effect of tapeworm infection was blocked by neutralization of IL-10. These findings indicate that, in a nonpermissive system, a viable infection of *H. diminuta* exerts a profound anticolitic effect, both prophylactically and as a treatment, which is mediated at least in part via IL-10 [54]. Similarly, Hunter et al. (2010) found *H. diminuta* to reduce the severity of DNBS-induced colitis in mice. The presence of the tapeworm in the small intestine was effective in increased intestinal expression of messenger RNA (mRNA) for CD14, arginase-1, and found in inflammatory zone 1 (FIZZ1)—markers of alternatively activated macrophages (AAMs). Anticolitic effect and the protection was lost when the intestinal macrophages were depleted using clodronate treatment. Moreover, procurement of AAMs from patients’ blood and adoptive transfer of these cells can rescue the protection. These results suggest that AAM cells play an important role in providing protection to DNBS-induced colitis. [55].

A study by Melon et al. (2010) found increased IL-4 and IL-10 production by spleen cells in mice infected with *H. diminuta* eight days prior to receiving DNBS compared to uninfected mice and uninfected mice with induced colitis. Furthermore, the infected mice were protected from DNBS-induced colitis [56]. Johnston et al. (2010) examined the impact of a phosphate-buffered saline-soluble high-molecular-mass fraction obtained from adult *H. diminuta* (HdHMW) on protection against colitis induced by DNBS. Authors observed reduced macrophage activation by HdHMW, which may have been partly caused by a mannosidase- and neuraminidase-insensitive glycoprotein. Mice cotreated with HdHMW displayed significantly less inflammatory disease, manifested as reduced TNF-alpha production (proinflammatory role) and increased IL-10 and IL-4 production (anti-inflammatory or tissue reparative roles). Results suggest that the tapeworm is a source of molecules whose immunomodulatory properties could lead to the development of new drugs. However, complete identification of anti-inflammatory and immunosuppressive particles is necessary [51].

Persaud et al. (2007) examined the immune response by male mice in response to *H. diminuta* and the role of CD4+ cells (predominantly T cells) in tapeworm expulsion and anti-colitic effects. Mesenteric lymph node cells stimulated by con-A from infected mice displayed elevated production of the Th-2 type cytokines IL-4, -5, -9, -10 and -13. Mice deficient in CD4 were not protected from DNBS-induced colitis by infection with *H. diminuta*. CD4+ cells appear to represent a key component of the observed anti-colitic effect. In addition, CD4 knock-out mice (CD4 −/−) were capable of expelling tapeworms despite displaying no response to infection. Infection with *H. diminuta* entailed protection from DNBS-induced colitis [57].

A study conducted by Matisz et al. (2015) found the application of dendritic cells (DCs) derived from bone marrow stimulated with an *H. diminuta* antigen (HD-DCs) successfully alleviated DNBS-induced colitis. No such protection was observed in IL-10 knockout mice or RAG-1 knockout mice [58]. In addition, HD-DCs were found to modify the anti-colitic CD4(+) T cells with the help of IL-10, and transfer of HD-DCs suppressed DNBS-induced colitis through activation of recipient IL-4 receptor-α [59]. This indicates the possibility of using properly stimulated and polarized cells for therapeutic purposes.

Matisz et al. (2018) also elucidated the mechanisms by which dendritic cells ameliorate the severity of DNBS-induced colitis in mice following exposure to antigens from *H. diminuta*. They note that HD-DCs can suppress colitis via an alternative MHC II-independent pathway that involves, in part, mobilization of T cell responses [60]. Reyes et al. (2016) report that IL-22 reduces the protection offered by *H. diminuta* from DNBS-induced colitis by inhibiting IL-25 and IL-10 expression [61]. Li et al. (2021) examined the effect of DNBS on the outcome of *H. diminuta* infection. In one experiment, mice were challenged with DNBS four days after tapeworm infection, and in the other, they were challenged three days earlier than infection. The results showed that infection protected against a severe course of DNBS-colitis and accelerated recovery after DNBS. In addition, DNBS-colitis did not affect the ability of the mouse to expel the tapeworm (mice expelled *H. diminuta* by 11 days post-infection) and to mount Th2 immune responses [62].

In contrast to mice, rats are the conventional definitive host of *H. diminuta* and are typically colonized without feeling ill effects, only responding with a mild type 2 immune response to infection. Jirků Pomajbíková et al. (2018) examined how *H. diminuta* infection at different life cycle stages protected against DNBS-induced colitis in rats. It was found that rats with immature tapeworms demonstrated increased IL-4, IL-13 and IL-10 expression but were not protected against colitis, while mature *H. diminuta* caused less severe clinical symptoms [63].

Wang et al. (2017) assessed whether triggering a memory response against *H. diminuta* inhibits DNBS-induced colitis in mice. The results suggest that mice infected with *H. diminuta* had less severe inflammation, and that secondary infection or antigen delivery protected from DNBS-induced colitis [64]. Reyes et al. (2015) found that splenic CD19(+) B cells from *H. diminuta*-infected mice exerted a protective function against DNBS-induced colitis by producing TGF-β [65].

Reardon et al. (2001) investigated how *H. diminuta* tapeworm infection can modulate murine colitis. Inflammation was induced in a mouse model by DSS added to drinking water; the mice were either infected with *H. diminuta* before exposure or afterwards. The mice infected with *H. diminuta* before DSS treatment had less severe disease than those infected after exposure, manifested as reduction of occult blood and normalization of epithelial ion transport; however, the latter also demonstrated reduced colitis symptoms. Reardon et al. (2001) also notes that *H. diminuta* infection given before or after exposure to DSS can ameliorate the epithelial ion transport irregularities observed in DSS colitis [66]. Intraperitoneal injection of HdAg (*H. diminuta* antigen) suppressed DSS-induced colitis and reduced IFN-γ, IL-17, TNF-α production and increased IL-10 concentration [67].

Infection with helminth parasites can ameliorate or exacerbate inflammatory diseases. Infection with *H. diminuta* ameliorated FCA-induced monoarthritis in mice. Injection of FCA into mice resulted in knee swelling, a decreased pain threshold, increased blood flow to the knee, and increased production of TNF-α and IL-12p40. *H. diminuta* infection also reduced the severity of arthritis in mice [68]. In contrast, Graepel et al. (2013) found that the immune response to *H. diminuta* infection exacerbated K/BxN serum-induced joint inflammation. Mice infected with *H. diminuta* had more severe disease, which correlated with greater C5a and mast cell activation. *H. diminuta* infection was also found to exacerbate K/BxN serum-induced polyarthritis in BALB/c mice [69].

McKay and Wallace (2009) found that *H. diminuta* infection did not protect rats from acetic acid-induced gastric ulceration. While the authors do not dismiss the possibility that infection with other helminths could be anti-ulcerogenic, helminth therapy as a treatment for inflammatory disease is likely to be both disease- and helminth-specific [70].

Fan et al. (2004) found that immunization with non-viable oncospheres of *H. diminuta* offered complete protection against infection by homologous parasites in rats or mice [71]. In turn, Arai et al. (2018) report that a crude worm extract of *H. diminuta* inducing an immune response in previously infected mice could be used to limit the severity of colitis, regardless of the age of the host [72]. Smyth et al. (2017) assessed the suitability of *H. diminuta* cysticercoids (HDC) as anti-inflammatory therapeutics. They found that the effectiveness of HDCs depended on the purpose of the therapy, with the effectiveness of treatment varying considerably between individuals. Nevertheless, HDCs were found to be effective against migraine headaches, Tourette’s syndrome, chronic fatigue syndrome, major depressive disorder and autoimmune diseases. Mixed results were obtained for allergies and arthritis [73].

Parasitic helminths do not always alleviate the histopathology and symptoms of colitis. Hunter et al. (2007) found that infection with *H. diminuta* caused a significant exacerbation of oxazolone-induced colitis [74]. In addition, Wang et al. (2010) observed that infection with *H. diminuta* may maximize oxazolone-induced colitis with higher IL-5 and eosinophil production [75]. Perhaps the different research results are due to the variety of methods implied to induce disease in animal models.

Increasing changes in the intestinal biota are being noted in Western society, resulting in an increased risk of developing autoimmune diseases and inflammation-related diseases, such as colitis-associated colon cancer. Sauer et al. (2021) examined how *H. diminuta* infection may influence biota development and hence, breast cancer using the C3(1)-TAg mouse model of breast cancer. The results indicated that tapeworm infection had no effect on the induction or progression of breast cancer [76]. However, it is not known how infection with a tapeworm, a yeshiva parasite, would affect the development of cancers of the small and large intestine. The above-mentioned immunological processes and the influence of the parasite on the oxidative balance could be reflected in the cancer processes of the intestinal tract.

The influence of *H. diminuta* on the absorption of heavy metals was also investigated. Tapeworms have the ability to absorb heavy metals (especially elements in excess) from the host body. Zn or Zn/Cd burden was estimated based on the levels of biochemical parameters (total protein, albumin, urea, glucose, triacylglycerols, non-esterified fatty acids, cholesterol, alkaline phosphatase, aspartate aminotransferase, uric acid and Mg, Ca, P and Zn). Infection was only found to influence total protein, urea and phosphorus levels. In the serum of uninfected rats, the zinc level did not change after overdosing zinc lactate and Zn/Cd hyperaccumulating plants. In turn, Zn/Cd hyperaccumulating plants reduced the level of zinc in the serum of rats infected with *H. diminuta*. On the other hand, an overdose of zinc lactate in infected rats did not decrease zinc levels in their blood. Authors suggested that the element form, which is received by host can affect zinc levels in the blood serum [77].

There is a lot of evidence of beneficial impacts of *H. diminuta* infections on the course of autoimmune diseases, especially colitis. This effect is primarily due to the immunomodulatory properties of specific parasite proteins. The accurate identification and characterization of the molecular mechanisms of action of this tapeworm’s proteins and the study of the possibility of alleviating inflammatory responses may be crucial for the development of new drugs against autoimmune diseases and allergies.

## 6. The Host–Parasite Relationship

*H. diminuta* is used as a model organism to study complex interactions between host and parasite. The cestode *H. diminuta* uses a variety of insects as its intermediate host, and rats and humans as definitive hosts. Under laboratory conditions, two genera of tenebrionid beetles are used as intermediate hosts for *H. diminuta*: *Tenebrio* and *Tribolium*. Vertebrate hosts are much less common due to ethical constraints. Studies on relationships in the parasite–host system are mainly related to the effect of infection, ecological interactions, behavioral studies and other dependencies such as body weight and reproductive potential.

Insect hosts have varying susceptibility to being infected by a parasite. Possibly the most significant factor affecting the effectiveness of infection in larval insects infected with parasitoids is host age, which correlates with body size [78]. It is also associated with the defensive capabilities of invertebrate hosts [79]. In addition, Shostak (2008) observed a general decline in beetle activity with age, which may reduce encounters and feeding on parasite eggs, thus reducing the levels of infection in older hosts [80].

An important consideration when using insects in host–parasite research is the accurate prediction of parasite load in experimentally-infected hosts in relation to the dose of infective stages. For *H. diminuta*, Dhakal et al. (2018) reports a higher proportion of established cysticercoids at concentrations above 100 eggs per 10 µL compared to lower concentrations [81].

The presence of a parasite can have a serious impact on many aspects of the host. Hitchen and Shostak (2009) report that *H. diminuta* induces changes in the expression of selected genes in the invertebrate host *T. confusum*; HSP68, a predicted sugar transporter, endoglin and a pheromone-binding protein were upregulated in infected beetles, while thaumarin-like proteins and prophenoloxidase 2/3 were downregulated. Hence, while changes in the expression of certain genes may be protective for the host, they can also favor the persistence of *H. diminuta* [82]. Infection with the parasite may therefore affect the gene expression of the host and affect its metabolism, immunology, physiology, etc. All the more, the selection of an appropriate invasive dose and host age in studies seems to be crucial for the repeatability and reliability of the data obtained.

Parasite infection is associated with stress, which can affect bodyweight and size, reproductive success, waste production or food intake. However, beetles infected with *H. diminuta* cysticercoids demonstrate no changes in body weight, frass production or food intake [83,84]. Infected *T. molitor* beetles show no preference for infective faeces from rats [85]. In turn, Ponton et al. (2011) reports that beetles infected by *H. diminuta* increased their total food intake, particularly carbohydrates, compared with uninfected insects. This increased intake was only observed during the first 12 days post infection, when the parasite grows and develops into a metacestode [86]. This seems logical considering the intensive development of the larva in the body of an insect. In turn, male *Gnatocerus cornutus* infected with *H. diminuta* had smaller mandibular horns (used in mating) and lower body mass [87]. As described below, these differences are reflected also in hormone levels in infected beetle males.

Infection with a parasite affects host reproduction. Such reduction in fertility can be an adaptive response that works to divert resources from host reproduction toward longevity, growth and/or immunity [88]. *T. molitor* beetles infected by *H. diminuta* demonstrated reduced fecundity in conditions of extreme host crowding and lower volumes of retained eggs in virgin females [89]. Also, in the other intermediate host, *T. confusum*, infection with *H. diminuta* resulted in a major reduction in fecundity, but did not significantly affect the size of the eggs [90]. Infected male *T. confusum* showed reduced mating vigor and consequently, inseminated fewer females compared to uninfected males [91]. In turn, Hurd and Ardin (2003) demonstrated that female *T. molitor* produced significantly more offspring after mating with infected males compared to uninfected males [92]. Moreover, infection resulted in an enlargement of the male accessory glands. Infection also increases oviposition by virgin and mated beetles 6–9 days post infection but does not affect the total number of eggs laid over 18 days [93]. The parasite also influenced fertility by acting on the juvenile hormone and vitellogenin.

Parasitic infection also not only affects reproductive success by changing egg size, but also by influencing attractiveness to the opposite sex. Worden et al. (2000) demonstrated that the odor of male *Tenebrio molitor* can indicate the intensity of *H. diminuta* infection to female beetles. The authors suggest that while the odors of all male *T. molitor* attract females, the parasitized males were less attracted to the females. Infection also has an impact on the number of offspring; females associated with infected males had fewer offspring than the ones associated with uninfected males [94].

Willis and Poulin (2000) showed that female rats spend more time investigating the urine of non-parasitized males than that of males infected with *H. diminuta*, and infected males had lower testosterone levels in their blood than non-infected males [95]. Infection by *H. diminuta* was also found to affect the choice of mate in a wild strain of *T. molitor*, with virgin female *T. molitor* spending more time near noninfected males and copulating more frequently with them than with infected males [96].

The *T. molitor*–*H. diminuta* relationship is also used to investigate learning behavior. Infection was found to influence the locomotion and the rate of learning in the beetles; the infected *T. molitor* moved and learned more slowly than the uninfected beetles, although they reached the same level of learning. Learning difficulties were visible mainly at the beginning of the experiment [97].

Studies of the behavioral alterations of tenebrionid beetles infected with *H. diminuta* found infected beetles to be more frequently exposed to contact with predators, i.e., not concealed under cartons, than uninfected beetles. However, both the infected and uninfected beetles were consumed equally willingly by predators. These results suggest that host behavioral changes do not facilitate transmission but were a consequence of parasite-induced pathologic changes [98].

As can be seen, parasite infestation can affect behavior, hormone levels, sexual characteristics and reproductive success in infected insects. However, little is known about the effect of infection on similar parameters in mammals. Below we present the possible influence of *H. diminuta* on the behavior of the definitive host; however, the influence of adult tapeworms on the hormonal balance and fertility/fecundity of definitive hosts remains unexplained

Infection with *H. diminuta* impacts exploratory behaviors in both beetles and rats. In a study of the effect of intestinal infection by *H. diminuta* on the behavior and functions of the CNS in rats, it was found that infection was associated with changes in behavior and impaired locomotor activity. However, infection positively influenced spatial memory and new object recognition, suggesting that the parasite–host interrelationship modified noradrenergic and serotonergic neurotransmission in the host [99].

As described above, the presence of *H. diminuta* in the definitive host organism results in various morphological, histological, immunological, physiological and biochemical changes. Goswami et al. (2011) showed that rats infected with *H. diminuta* had reduced hemoglobin values. Moreover, in intestinal mucosa, there were observed pressure atrophy, more compressed and atrophied villi, degeneration and desquamation of lining epithelium cells and excessive mucin secretion in the lumina [100]. Kosik-Bogacka and Kolasa (2012) report that infection with *H. diminuta* caused a reduction in villus length and greater crypt deepening in the duodenum and jejunum [101]. Ahmad et al. (2015) noted that *H. diminuta* infection was associated with multiple mucosal ulcers and absences of intestinal villi from the surface epithelium [102]. Infection of rats with *H. diminuta* is accompanied by changes in ion transport in the epithelium of the colon, indicating that hymenolepiasis inhibits sodium and chloride ion transport [103]. Also, infection with *H. diminuta* reduced ion transport in the ileum of rats [104]. In turn, Zimmerman et al. (2008) showed that *H. diminuta* can affect the absorption process in the intestine while altering intestinal motility and slowing down intestinal processes; the parasite released 3′,5′-cyclic guanosine monophosphate (cGMP) to the intestinal lumen and altered the physiology of the host digestive process [105].

In addition, infections with *H. diminuta* lead to changes in enteric cholinergic activity [24]. Kapczuk et al. (2022) examined the influence of *H. diminuta* infection on apoptosis in the small and large intestines based on pro-apoptotic and anti-apoptotic gene expression. Hymenolepiasis was found to enhance apoptosis in the small and large intestine of the host by increasing pro-apoptotic *Bax* gene expression while decreasing anti-apoptotic *Bcl-2* gene expression and caspase cascade activation. Hence, apoptosis appears an important process both for the survival of the parasite and the host defense mechanism [106].

The aforementioned data indicate that tapeworm infection can have negative effects, leading to changes in the functioning and environment of the intestine. This, in turn, may be reflected in the composition and structure of the intestinal microbiota.

The gut microbiota is very important to host health and there is hence a need to understand the mechanisms influencing its composition and diversity. Several studies have suggested that intestinal parasites, including helminths, interact with the bacterial flora [107,108]. McKenney et al. (2015) investigated the changes that occur in the microbiome of the caecum of rats after *H. diminuta* infection. Infection led to a shift from *Bacilli* to *Clostridia* species; a higher contribution of *Bacilli* to the microbiome is associated with a Western diet characterized by supplying sugars and fat, while *Clostridia* are known to decrease the likelihood of allergy. These changes suggest the potential mechanisms by which this helminth might exert therapeutic effects [109].

Recent reports also indicate that intestinal helminths may influence the composition of the microbiome and its impact on the functioning of the immune system. Wegener Parfrey et al. (2017) observed minor changes in microbiota community composition during *H. diminuta* infection in rats. However, these compositional changes appear to be minor. These results suggest that *H. diminuta* could be used as a therapeutic helminth that can safely manipulate the mammalian immune system without disrupting the rest of the gut microbiota [39]. In turn, Shute et al. (2020) found that PBS-soluble extract of adult *H. diminuta* promoted the growth of anaerobic bacteria on M2GSC agar and increased mucosa-associated and fecal bacterial communities, characterized by increased *Lachnospiraceae* and *Clostridium* cluster XIVa, in mice infected with *H. diminuta*. Thus, infection with *H. diminuta* results in subtle but distinct changes to the colonic microbiota [110]. Shute et al. (2021) proposes that the gut microbiome is a critical component of the anticolitic effect of *H. diminuta* therapy and suggest that the anti-inflammatory effect of a parasitic infection may be partly due to the presence of intestinal microflora [111].

Also, intestinal parasites can be affected by the insect gut microbiome. Fredensborg et al. (2020) used *T. molitor* to explore interactions between the infection status of *H. diminuta* and gut microbiota. The results indicate that infection status and beetle age significantly influence the gut bacteriome and mycobiome. A detailed examination showed the infected beetles to have a higher abundance of proteobacteria in the gut with a less abundant but more diverse mycobiome compared to non-infected beetles. In addition, administration of tetracycline resulted in reduced parasite establishment compared with untreated infected beetles. The observed differences indicated that the host microbiome may greatly influence the hatching of *H. diminuta* eggs [112].

Interactions between the parasite and the microbiome may be important in the immune response to infection and the influence of the parasite on the behavior of the host. Changes in the microbiome under the influence of the parasite may affect the physiology of the intestine, and thus a number of biochemical and physiological processes related to the gastrointestinal tract and the enteric nervous system.

Before *H. diminuta* cysticercoids enter the body of the definitive host, they produce at least 70 potentially antigenic proteins which are encountered by the mammal host soon after infection. Most of the identified proteins are structural proteins (actin, myosin, tubulin), heat shock proteins and those involved in metabolic processes. Although many were previously known to be antigens or to be involved in host immune evasion or immune modulation in other helminths, some were identified for the first time as antigenic proteins. Hence, *H. diminuta* cysticercoids appear to be endowed with proteins with antigenic properties, which play key roles in host–parasite interactions and help the tapeworms survive during the early stage of invasion [113]. These proteins may be involved in the early stages of invasion and contribute to successful colonization. On the other hand, they may represent multifunctional proteins with immunomodulatory properties. Understanding their role requires the use of in vitro studies using recombinant proteins.

Sulima et al. (2018) compared the proteomic profiles of adult and cysticercoid *H. diminuta*. The authors identified certain stage-specific and common proteins as known antigens and immunomodulators. The identified proteins were engaged in mechanisms crucial for parasite invasion, establishing infection and escaping from host immune defenses. Many molecules were found to be involved in host–parasite interactions in both developmental stages [114]. The indicated differences may reflect physiological differences in individual developmental stages. These proteins may be related to the changes in biochemistry, receptor expression and microbiome in the host gut described previously.

As many proteins present on the cell surface play key roles in successful invasion [114], Młocicki et al. (2018) analyzed the surface proteins (surfaceomics) and immunoproteome of the adult tapeworm and identified a number of new immunogenic proteins involved in key metabolic processes that also played a role in parasite–host interactions. Many proteins were identified as potentially antigenic, indicating that the immune response of the host may be stimulated by various mechanisms, including those which activate protein export via unknown pathways (e.g., structural proteins). Their proteomic analysis demonstrated that the somatic and surface proteins were able to promote immunomodulatory functions and mediate parasite–host interactions [115].

Mazanec et al. (2021) proved the presence of extracellular vesicles (EVs) in adult *H. diminuta*. A proteomic analysis showed that the purified extracellular vesicles contained proteins originating from both *H. diminuta* and the host. These results indicate that EVs play an important role not only in parasite–host communication, but also in the manipulation of the host organism by the parasite [116].

*H. diminuta*-intermediate host and *H. diminuta*-definitive host studies can explain the influence of the parasite on a number of key metabolic, immunological, behavioral, hormonal processes, microbiome and many others, crucial for the functioning of the host and the evolutionary success of the parasite. The above studies indicate that the parasites have developed a number of adaptations and strategies to ensure their survival during the coevolution with the host. Many of these have direct effects on the host, its behavior and physiology. There is increasing evidence that these processes are interconnected and much more complex than one might think.

## 7. Treatments

The most common definitive hosts of *H. diminuta* are synanthropic rats [117]. However, humans, especially children, can also accidentally enter the life cycle [118,119,120]. *H. diminuta* infections are asymptomatic, but there may be abdominal pain, diarrhea and irritability [119]. So far, 1561 published records of infection with *H. diminuta* from 80 countries have been identified [121]. The drugs of choice for treating *H. diminuta* infestations are praziquantel and niclosamide [122]; however, there is growing interest in identifying substances with natural anthelmintic properties.

In many regions of the world, the natural anticestodal properties of plants have been employed to fight against parasites. The leaf extract of *Strobilanthes discolor* is used by the Naga tribes of north-eastern India in the treatment of intestinal worm infection. Tangpu et al. (2006) examined the anti-cestode potential of this extract in rats infected by *H. diminuta* [123]; it was found to reduce adult worm numbers and demonstrate notable efficacy against larval stages with a similar efficacy against adult stages as praziquantel. Similar properties have been described for other plants: *Adhatoda vasica* [124], *Zanthoxylum rhetsa* DC [125], *Houttuynia cordata* [126], *Solanum myriacanthym* Dunal [127], *Cassia alata* [128], *Carex baccans* [129], *Oroxylum indicum* [130], *Caesalpinia bonducella* [131] and *Acorus calamus* [132].

Another popular plant genus among antihelminth researchers is *Senna*. Notably, *Senna occidentalis* was found to demonstrate significant potential anthelmintic properties [133]; *S. occidentalis* extract caused severe damage throughout the body tegument of *H. diminuta.* In addition, *Senna alexandrina* demonstrates strong antimicrobial activity, inducing ultrastructural changes in *H. diminuta* including destruction of tegument, vacuolization of the syncytium and depletion of the parenchymatous layer [134].

Ukil et al. (2018) report that *S. alata*, *S. alexandrina* and *S. occidentalis* altered morphology, ionic concentration and neurotransmission in a study of mitochondrion ultrastructure in *H. diminuta* [135]. They also note that the leaf extracts of the three plant species decreased the activity of mitochondrial enzymes, resulting in malfunction and cell death. These findings underline the chemotherapeutic properties of *Senna* [136].

Roy et al. (2020) found the ethanol leaf extracts of *S. alata*, *S. alexandrina* and *S. occidentalis* to generate ROS and induce apoptosis in *H. diminuta*. Hence, the leaf extract may have value as an anthelmintic agent [137].

Coconut also appears to be useful. Although chloroform extract of coconut (*Cocos nucifera*) was not enough to kill *H. diminuta* at a dose of 0.1 g/kg body weight/day, a dose of at least 0.4 g/kg was found to stop egg excretion and kill adult worms in vivo [138]. In addition, papaya seeds have yielded satisfactory results against *H. diminuta* infections in rats; more than a 96% reduction in eggs per gram and 90% anthelmintic efficacy were observed after 21 days of treatment [139].

In addition, the components of plant extracts can also exhibit anthelmintic properties. For example, extracts from pineapples, papayas and figs contain cysteine proteinases (CPs) which have been systematically tested against intestinal nematode parasites [140]. Mansur et al. (2014) demonstrated that plant derived CPs are detrimental to *H. diminuta* maintained in temporary in vitro assays [10]. In addition, condensed tannins from pine bark extract, hazelnut pericarp and white clover flowers have demonstrated anthelmintic properties [141].

It is no secret that nature, including plants and mushrooms, can be a rich source of substances with therapeutic potential. Many of them have been used for hundreds of years by local communities. However, their effectiveness must be confirmed in laboratory conditions and in effective therapeutic doses, which, even if effective, may not always be safe. In addition, an active substance that could be obtained synthetically should be identified. Moreover, many of the above results were obtained in vitro and require confirmation of their effectiveness in vivo.

Other alternative treatments are being researched. Merwad et al. (2011) examined the efficacy of high hydrostatic pressure processing (HPP) on the viability of *H. diminuta* eggs based on egg hatch assay. HPP was found to effectively inactivate tapeworm eggs [142].

By using the tapeworm as a model organism, it was possible to evaluate the therapeutic properties of many plant species and their effectiveness against *H. diminuta*. Traditional medicine based on plants or extracts of plants is sometimes the main and only source of medical aid in developing countries. Therefore, there is a need to confirm antiparasitic potential of traditional medicine to find widely available, low cost and effective treatments against parasites.

## 8. Conclusions

*Hymenolepis diminuta* is a suitable model species in a wide range of modern and classical experimental parasitology research. It has been mentioned as an experimental model in nearly 140 research papers published since the year 2000 (Figure 3). As indicated in our review, this tapeworm represents a safe and important model for studying the molecular biology, biochemistry and immunology of tapeworm infections, with the rat tapeworm model being the most common base for studying host–parasite interactions. A considerable amount of research has attempted to identify possible treatments and evaluated the use of *H. diminuta* as a therapeutic agent itself.

The parasite affects a number of processes occurring in the host organism, demonstrates adaptations to changing conditions and interacts with the host by modulating its functioning and influencing its intestinal microbiota. This review systematizes current knowledge about *H. diminuta*, thus improving the general understanding of the biology of tapeworms and their adaptation to parasitism; this knowledge includes the complex interrelationship between the host and *H. diminuta*, as well as with other cestode species.

The variety of research carried out with the use of *H. diminuta* as a model highlights its importance in modern experimental parasitology and has been a valuable source of information regarding tapeworm biology, and their possible practical application in human and veterinary medicine. Even so, there are still many aspects to be explored and many questions that need to be answered by *H. diminuta*.

## Figures and Tables

**Figure 1 pathogens-11-01439-f001:**
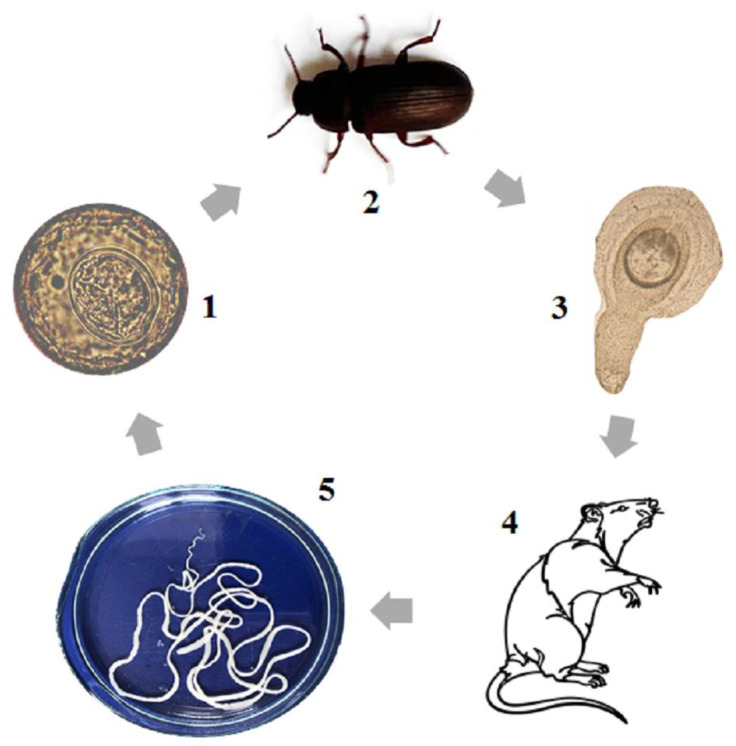
The lifecycle of *H. diminuta* (1—egg with oncosphere, 2—the intermediate host (beetle), 3—cysticercoid, 4—definitive host (rat), 5—adult stage).

**Figure 2 pathogens-11-01439-f002:**
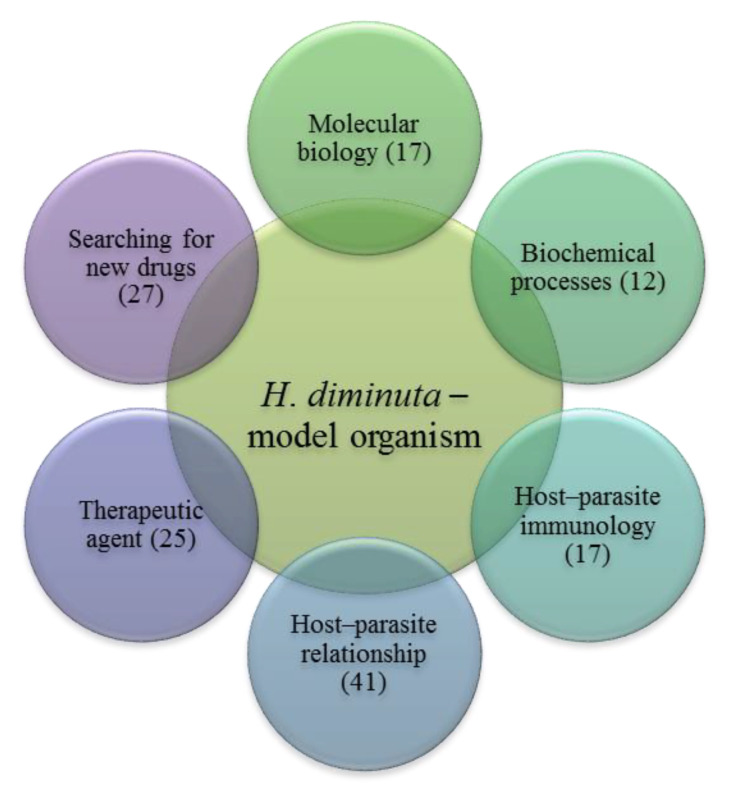
The use of model species *H. diminuta* in scientific research. The number of publications in particular groups is given in parentheses.

**Figure 3 pathogens-11-01439-f003:**
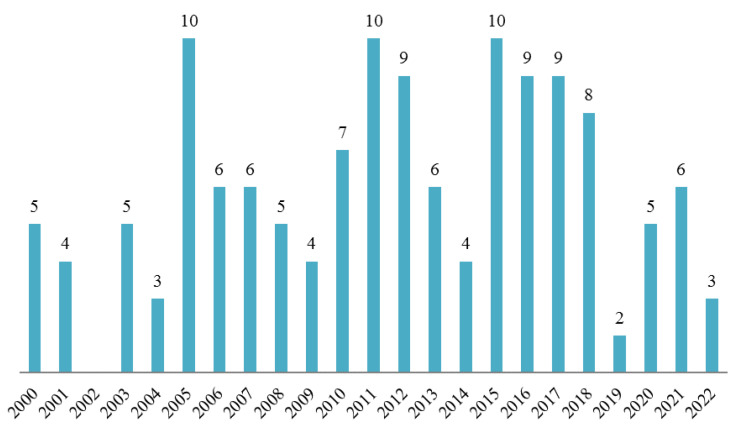
The number of experimental publications in the years 2000–2022 using *H. diminuta* as the model organism.

## Data Availability

Not applicable.

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
