# Peer review of "The Tapeworm Hymenolepis diminuta as an Important Model Organism in the Experimental Parasitology of the 21st Century"

_pathogens, 2022, doi:10.3390/pathogens11121439_

Round 1

Reviewer 1 Report

The review in the title: The tapeworm Hymenolepis diminuta as an important model organism in the experimental parasitology of the 21st century. I really enjoy reading this review. I have only one comment on line 142; TLR2 no need to use italic front.

Author Response

Thank you for the review. All changes have been included in the manuscript.

Reviewer 2 Report

The authors describe Hymenolepis diminuta as an important model species in studies of cestodiasis, including the search for new drugs, treatments, diagnostics, and biochemical processes, as well as its host-parasite interrelationships.

In general, the approach is comprehensive, well-organized, and well-written.  However, a few clarifications and corrections are necessary, as listed below: 

Figure 2 caption:  these are parentheses , not brackets

Line 232:  ameliorate, not ameliorating

Line 240: “The host immune system is designed to protect against foreign objects, including parasites, and hence affects parasite abundance and survival.” 

Instead of “objects”, use organisms

Line 247: Repetition of previous paragraph.   Infection with a helminth typically evokes a Th2 cell response in a mammalian host. Omit.

Line 248: In a murine host, the response follows the classical Th2 pattern associated with helminthic 248 infections and is dominated by IL-4, IL-5 and IL-13 [7].  Murine can refer to any rodent; use mouse instead of murine

Line 311: define ROS

Line 394: missing hyphen: tnf-alpha

Line 402: do not italicize helminth therapy

Line 417:  missing closed parentheses: Wang et al. (20100

Line 426:  I don’t understand this statement:

H. diminuta has also been found to protect the host organism from the harmful effects of some heavy metals and to absorb zinc in the tissue/body. Zn or Zn/Cd burden was estimated based on the levels of total protein, albumin, urea, glucose, triacylglycerols, non-esterified fatty acids, cholesterol, alkaline phosphatase, aspartate aminotransferase, uric acid, and Mg, Ca, P and Zn. Infection was only found to influence total protein, urea and phosphorus levels.

If it only affects Mg, Ca, P levels, how does H. d. protect against Zn and Cadmium?  Also, Zinc is often deficient in malnourished populations, so decreasing zinc could be detrimental to health.    

Line 440: “Insect hosts have varying ability to be infect a parasite.”  Do you mean: Insect hosts have varying ability to be infected by a parasite?  Or better: Insect hosts have varying susceptibility to being infected by a parasite?

Line 485:  I am confused by this statement:

“The authors suggest that while the odours of all male T. molitor attract females, the parasitized males were less attracted to the females.” 

Are the parasitized males less attractive to the females, or do the parasitized males have decreased sex drives?

Line 496: “Infection was found to influence the locomotion and the rate of learning in the beetles: the…”  Should be a semi-colon, not a colon

Line 504:  “ …. parasite-induced pathology [98]. “

Pathology is the study of the study of the causes and effects of disease; you mean parasite-induced pathologic changes  

Line 506:  Missing a word after exploratory (presumably behaviors):

Infection with H. diminuta impacts exploratory ---------- in both beetles and rats.

Line 514:  explain: “pressure atrophy”

Line 625:  S. alata, S. alexandrina and S. occidentalis.   Should be: Se. alata, etc.

Line 626:  ROS not defined

Line 663: “Even so, there are still many ways to be explored and many”

Better: Even so, there are still many aspects

Author Response

Thank you for the review and suggestions. All editorial changes have been incorporated in the manuscript. Clarifications regarding the following suggested corrections are included below:

Line 485 (line 592 in new version of manuscript): The parasitized males are less attractive to the females.

Line 625 (line 750 in new version of manuscript): The authors (Ukil et al., 2022; Roy et al., 2016; Roy et al., 2020) use the abbreviation S. alata, S. alexandrina and S. occidentalis.

Reviewer 3 Report

The review describes tapeworm H. diminuta role as a model organism for studying various aspects of immunological, parasitological, and clinical applications. Overall, the authors have covered relevant literature and provided a comprehensive review of this model organism. I have the following recommendations/comments:

1-    This is a general comment: Throughout the text, the authors tend to write the results as it is without well describing the findings and without providing a reason why that finding is important. It is very important to provide a perspective and some sort of conclusion to better understand the message that the authors are trying to convey.

2-    H. diminuta is an excellent model tapeworm in IBD research in laboratory mice for studying DSS/TNBS-induced colitis. The authors should first describe how  H. diminuta infection affects normal laboratory mice before describing its use as a treatment model in colitis. It is not clear  in the absence of an intermediate host (beetle) how long the infection last in the mice in terms of days/months. Or whether DSS/ DNBS-induced colitis promotes the persistence of parasites in the gut when the epithelial barrier integrity is compromised.

3-    On page no 8 line 330 the findings of Hunter et al (2010) are incorrectly described. There are three major points, first  H. diminuta protects DNBS -induced colitis in mice. Second, the protection is lost when the intestinal macrophages are depleted using  clodronate treatment. Third, the adoptive transfer of AAM can rescue the protection. All these findings imply an important role of AAM in providing protection to DNBS -induced colitis when using  H. diminuta as a model tapeworm infection. The authors should correct it.

4-    Line 355, please correct the typo, it looks like the authors are indicating two KO; IL-10  and rag-1 KO?

5-    Authors have indicated HdHMW (high molecular mass fraction)  H. diminuta. Please describe it well. What it contains? Does it contain live tapeworm or just the secretory products tapeworm.

6-    Please introduce the term cestodology before using it in the second paragraph.

Author Response

Thank you for all the suggestions and valuable comments improving our review. All changes are included in the corrected manuscript and presented as “track changes”. We added necessary comments along the manuscript, to indicate the importance and perspectives of the presented research. Thanks to the suggested changes, this work is more complete. We hope that the changes introduced are satisfactory.

Clarifications regarding the suggested corrections are presented below:

1 – Moreover, at the end of each chapter, a paragraph summarizing the presented research has been added.

2 - Influence of H. diminuta infection on normal laboratory mice has been described in the line 379, and effect of DSS/ DNBS-induced colitis to the persistence of parasites in the gut in the line 445.

3 - The findings of Hunter et al (2010) have been corrected – please see the line 392 in new version of manuscript.

4 – The typo was used - "of" has been corrected to "or". We have described two types of KO mice.

5 – The results of research conducted by Johnston et al. (2010) have been re-described in the line 408.

6 – The term cestodology is defined in the line 34 of corrected version.

Round 2

Reviewer 3 Report

The authors have satisfactorily addressed all the concerns.